# Back-Carrying in Children Is Related to Lower Limb Development [note 1]

**DOI:** 10.3390/children9020263

**Published:** 2022-02-15

**Authors:** Mariaan van Aswegen, Stanisław H. Czyż, Sarah J. Moss, Francois Steffens

**Affiliations:** 1Physical Activity, Sport and Recreation (PhASRec), Faculty of Health Sciences, North-West University (NWU), Potchefstroom 2520, South Africa; 20383800@nwu.ac.za (M.v.A.); Hanlie.Moss@nwu.ac.za (S.J.M.); 2Department of Motor Skills, Faculty of Physical Education and Sport, Wroclaw University of Health and Sport Sciences, 51-612 Wrocław, Poland; 3Incubator of Kinanthropology Research, Faculty of Sport Studies, Masaryk University, 62500 Brno, Czech Republic; 4Department of Consumer and Food Sciences, University of Pretoria, Pretoria 0002, South Africa; steffensfe2@gmail.com

**Keywords:** back-carrying, lower limb development, tibiofemoral angle, Setswana children

## Abstract

Back-carrying of children is a culturally accepted method of transport and safekeeping of babies in many cultures. Developmental consequences related to back-carrying practices have not been directly investigated. This study determined the relationship between frontal and transverse plane lower limb (LL) development, and back-carrying practices, in black Setswana-speaking children. In 691 2- to 9-year-old Setswana-speaking children, the tibiofemoral angle, intermalleolar distance, femoral anteversion angle (AVA) and tibial torsion angle (TTA), were measured to determine LL development. Back-carrying practices were recorded with a questionnaire and Classification and Regression Tree (CART) was used for the analyses. Significant (*p* < 0.001) relationships, between back-carrying practices and LL development, were discovered. Statistically significant greater genu valgum (F(5, 690) = 7.2, *p* < 0.001), greater internal TTAs (F(9, 684) = 17.8, *p* < 0.001), and smaller AVAs (F(13, 685) = 5.1, *p* < 0.001) were observed in children back-carried more frequently than children back-carried less frequently. There are relationships between back-carrying practices and LL development in both the frontal and transverse plane. However, the genu valgum, internal TTA and smaller AVA noted in more frequently back-carried children is still within normal limits, thus no educational intervention in back-carrying methods or durations is required. Further research should determine the exact back-carrying practice factors (age until which the child is back-carried) impacting lower limb development the greatest.

## 1. Introduction

South African children of black ethnicity are generally back-carried (BC) by their mothers, with the aid of blankets or towels [1,2,3]. The child’s legs are either placed spread around the mother’s back or left hanging down on the mother’s back, while the blanket or towel envelops the child’s back and buttocks and is tied at the mother’s front. The latter method (legs hanging down) is usually utilised in early infancy. The tradition of back-carrying (BC) is well accepted and widely practised by the mothers [1,4,5]. Mothers from a lower socio-economic status will back-carry (BC) their child for extended periods, as long as eight (intermittent) hours per day, while working or travelling on foot, to keep the child safe [1,3].

Child development can be disadvantaged and benefitted by back-carrying [3]. BC restricts head, upper and lower limb (LL) movements and limits crawling of the child during the sensorimotor developmental phase, leading to inhibition of vision [1,2,3]. The inhibited vision in BC children in turn contributes to hampered school readiness, fine motor skills and an inability to integrate movement with observational cues [3]. BC children develop strong emotional and social bonds with their mothers; however, some experience difficulties with unfamiliar situations, which was linked to their limited exploration ability while being BC for long periods [3]. Possible prevention of developmental dysplasia of the hip (DDH) in BC children, or contribution thereof to the rarity of DDH, is the most significant benefit [2,6,7,8]. The incidence of DDH in black children residing in Indiana, USA, was slightly higher than expected, with 4% of black children diagnosed with DDH, which may be attributed to the fact that these children are more likely transported in prams, than being BC [9].

DDH is treated with a brace (Pavlik harness) or hip spica cast that maintains hip flexion and abduction [7], resembling the BC position. Hence, researchers postulate that BC practices of black African mothers contribute to the low incidence of DDH in African populations [6,7,8]. In a retrospective review that included a total of 40,683 children under the age of 16 years, no infants presented with or were treated for symptomatic DDH [2]. Researchers concluded that BC of infants contributed to the low incidence of DDH found in Malawi [2]. These very significant results from Graham and colleagues [2] raise the question of whether BC would also have additional effects on LL development at the knee and ankle.

Normal angular alignment of the LL may be perturbed by both intrinsic and extrinsic factors, such as dietary and vitamin intake, physical trauma, infections, and high intakes of exogenous metals such as lead and fluoride which could all contribute to genu varum (bowlegs) [10]. Environmental factors such as the chair, desk, and bed the child uses could also affect postural alignment [11], and arguably BC could also affect LL alignment based on the postulated relationship between BC and DDH prevention [2,6,7,8]. One study reported a constant knee-valgus and external torsion pressure is produced on the LL and feet by the blanket securing the child to their mother’s back, when a child is BC with their legs spread (BC-LS) around the mother’s back [4]. In contrast, previous studies never investigated but postulated that long-duration BC-LS (or double diapering) leads to the development of genu varum based on the constant hip abduction, flexion and knee flexion position of the BC position [12,13].

Treatment of excessive LL malalignments such as genu varum poses potential high burdens on health systems [14]. Should BC contribute to the development of detrimental LL malalignments at the hip, knee (e.g., excessive genu varum or valgum) and ankle (increased tibial torsion), this financial burden could arguably be reduced or prevented through the education of mothers who frequently BC their children. Social or cultural causes of knee malalignments have not been directly investigated. The consequences of children BC in their childhood, in relation to their knee development have not been reported. Previous studies on the low incidence of DDH in BC children did not directly investigate the developmental consequences of BC practices [2,6]. The current study divulges important information about the relationship between LL developmental angle measures and BC Setswana-speaking children. Previous research indicated that BC is very common in black Setswana-speaking mothers [15]. LL development at the hip, knee and ankle of Setswana-speaking children followed similar general developmental patterns as reported for children of other ethnicities [14,16,17,18,19,20,21,22], regardless of BC practices [15]. The purpose of this study is therefore to determine relationships between LL angle measures and BC practices in Setswana-speaking children. Firstly, the relationship between frontal plane (FP) measures (tibiofemoral angle (TFA), intermalleolar (IMD) or intercondylar distance (ICD), and quadriceps angle (Q-angle)), in BC and non-back-carried (non-BC) children is determined. Secondly, the relationship between transverse plane (TP) torsion measures, femoral anteversion angle (AVA) and tibial torsion angle (TTA) in BC and non-BC children is determined.

## 2. Materials and Methods

This cross-sectional design study formed part of a larger study (Van Aswegen et al., 2020) [15] and was approved by the Human Health Research Ethics Committee (HREC) of the North-West University (NWU) ((NWU-00094-17-A1) and upheld the declaration of Helsinki of 1964. The data were obtained within the larger project, i.e., Lower Limb Development and Gait Kinematics of Back-Carried Setswana Children (see Appendix A).

### 2.1. Participants

All primary schools and crèches situated in Potchefstroom and Ikageng, listed by the departments of education and social development, were invited to participate voluntarily. Thirteen schools and crèches agreed to participate. Participants could only be recruited from three crèches, due to no black 2- to 9-year-old Setswana-speaking children in these schools. A total of 724 black Setswana-speaking children were measured and the results from 33 children with known musculoskeletal disorders (which may have caused malalignments of their LL) were excluded. Data from non-natively Setswana-speaking children were excluded from the analyses. Measurements from 691 children were included in the statistical analysis. Exclusion criteria were set to limit interference in the data and to allow tracking the development of the TFA in apparently healthy, black Setswana-speaking children [17,18,19]. Participants were grouped according to chronological age, rounded off to the nearest integer. Random sampling was not possible due to a limited number of schools in Potchefstroom agreeing to participate, and the smaller number of black Setswana-speaking children attending these schools as compared to those in Ikageng.

### 2.2. Procedure

The respective parent or legal guardian signed informed consent, and the child provided written or verbal assent, prior to the measurements. Upon inclusion, each participant was assigned a numerical identification number, which was written down on all their documentation (measurement proforma, questionnaire and informed consent), to allow linking of their clinical and questionnaire data. Information from a self-reported questionnaire was used to determine participant inclusion or exclusion [15].

The questionnaire collected data on the child’s development age, place of birth and commencement of independent walking age. Parents/legal guardians indicated whether leg deformities (bowlegs or knock-knees) were observed in the child and if these deformities changed (from bowleg to knock-knee), worsened or improved. Furthermore, parents indicated whether the child is/was BC, and how the child was BC, with the legs spread (BC-LS) around the mother’s back or BC with the legs hanging down. If parents did BC, they indicated the frequency of BC on a Likert scale from “very little”, “sometimes”, “often” and “always”. For the BC-frequencies, parents were asked where the infant or toddler was placed for safekeeping when they were with them while they were working or travelling, if their only option for safekeeping was BC, mothers frequently selected “always”. Parents also indicated pram use and swaddling.

The same researcher (MvA) performed all measurements once anatomic landmarks were identified through palpation and marked. Trained assistants maintained neutral hip rotation and full knee extension of the participants at measurement. Previously published measurement methods for the FP and TP measures were used [15,18,20,23], with varus angles reported as positive degrees (or centimetres for ICD) and valgus angles as negative degrees (or centimetres for IMD). TP measurements included the AVA and TTA, both measured in prone, with the knee flexed to 90° [15].

### 2.3. Statistical Analysis

Although sample size data estimates indicated a minimum of 63 participants per group (BC versus non-BC), data from 375 participants were available for analyses from the overarching larger study. All data were analysed with SPSS, Version 24, Inc., Chicago, IL, USA. The means of the IMD and the bilateral TFAs, Q-angles, TTA and AVA were determined from two to three measurements obtained per limb per variable, for each participant. The mean values of the various measurements were linked to each participant’s questionnaire data, indicating their BC preferences and frequencies. Participants with missing BC data were excluded from the analysis.

A Classification and Regression Tree (CART) was applied for the analyses, which allows the examination of multiple complex variables, whereby the algorithm focuses on the most relevant parameters [24]. The CART groups categorise independent variables and do not assume linear relationships or any underlying distribution. Instead, it aims to classify the sample from the root node (containing the whole sample’s means) into subgroups (nodes, intermediary nodes and terminal nodes) with the means of the subgroups as diverse as possible. The terminal nodes are the final subgroups which were applied for the statistical comparison. CART analysis was applied to each clinical measurement separately. The interaction between BC (both strategies: BC-LS and the legs hanging down) and age, and the TFA, Q-angle, AVA and TTA were first analysed, followed by an analysis of TFA, Q-angle, AVA and TTA in terms of age, sex and body mass index (BMI). CART analysis is exploratory and does not provide quantitative (e.g., *p* = value or test values) measures. Therefore, ANOVA was applied to determine significance between terminal nodes and partial eta squared as a measure of effect size. Duncan’s and Sheffe’s post hoc tests were used to screen for type I errors. The CART was used to separate the data of participants into groups of BC versus non-BC or in terms of BC frequency for example, groups of participants who were BC always, or less than always, per age.

## 3. Results

### 3.1. General Participant Descriptive Data

The FP and TP mean measures and standard deviations (SD) along with participant numbers are summarised in Table 1. From the total sample of 691 measures, the total number of children per age group and reported BC and BC-LS, non-BC and non-BC-LS with their respective unspecified numbers in the different age groups were as follows: 2-year-olds (N = 38; BC: *n* = 8, non-BC: *n* = 4, unspecified: *n* = 26; BC-LS: *n* = 17, non-BC-LS: *n* = 8, unspecified: *n* = 13), 3-year-olds (N = 52; BC: *n* = 13, non-BC: *n* = 3, unspecified: *n* = 36, BC-LS: *n* = 15, non-BC-LS: *n* = 11, unspecified: *n* = 26), 4-year-olds (N = 54; BC: *n* = 12, non-BC: *n* = 4, unspecified: *n* = 38, BC-LS: *n* = 11, non-BC-LS: 17, unspecified: *n* = 26), 5-year-olds (N = 112; BC: *n* = 19, non-BC: *n* = 11, unspecified: *n* = 82; BC-LS: *n* = 26, non-BC-LS: *n* = 32, unspecified: *n* = 54), 6-year-olds (N = 113, BC: *n* = 24, non-BC: *n* = 9, unspecified: *n* = 80, BC-LS: *n* = 38, non-BC-LS: *n* = 39, unspecified: *n* = 36), 7-year-olds (N = 98, BC: *n* = 19, non-BC: *n* = 6, unspecified: *n* = 73, BC-LS: *n* = 31, non-BC-LS: *n* = 42, unspecified: *n* = 25), 8-year-olds (N = 127, BC: *n* = 21, non-BC: *n* = 12, unspecified = 94; BC-LS: *n*-25, non-BC-LS: *n* = 70, unspecified: *n* = 32) and 9-year-olds (N = 97, BC: *n* = 20, non-BC: *n* = 9, unspecified: *n* = 68; BC-LS: *n* = 22, non-BC-LS: *n* = 53, unspecified: *n* = 22).

### 3.2. Frontal Plane Measures: Tibiofemoral Angle, Intermalleolar Distance, and Quadriceps Angle

The terminal CART nodes for TFA, BC-LS, and age are presented in Figure 1. Node 5 shows the smallest valgus mean for 2-year-olds (*n* = 38, TFA mean = −3.39° ± 3.43°, η^2^ = 0.022). In the 5- to 9-year-olds, node 6 shows the second smallest valgus mean and represents a larger nodal percentage of the participants (*n* = 457, TFA mean = −4.44° ± 1.83°, η^2^ = 0.008). Node 9 shows the largest valgus mean (*n* = 26, TFA mean = −5.98° ± 2.14, η^2^ = 0.002), representing 3-year-old children who were always BC. This CART was significant, as shown with ANOVA (F(5, 690) = 7.215, *p* < 0.001). Significant Sheffe post hoc tests were between nodes 5 and 7 (*p* = 0.016), nodes 5 and 9 (*p* < 0.001), nodes 5 and 10 (*p* = 0.009) and nodes 6 and 9 (*p* = 0.011). In summary, more frequent BC contributed to greater genu valgum (greater TFAs).

The analysis of the valgus IMD is presented in Figure 2, where the smallest valgus IMD was found in 6- to 9-year-olds (node 10), BC between often and always (*n* = 320, IMD mean = −0.72 cm, η^2^ = 0.000), whilst the second smallest valgus distance was observed in 9-year-old children (node 22) who were reportedly always BC (*n* = 22, IMD mean = −0.85 cm, η^2^ = 0.000) (Figure 2). The largest valgus IMDs were observed in 2- to 5-year-olds (node 4), who were BC-LS for specified frequencies (as opposed to unspecified frequencies) (*n* = 35, IMD mean = −2.98 cm, η^2^ = 0.048) and a significant ANOVA (F(13, 689) = 17.685, *p* < 0.001). Significant post hoc differences were observed for node 4 versus 10, 18, 21, 22, 25 and 26, node 10 versus 13 and 24 and node 13 versus 18. In summary, more frequent BC contributed to greater genu valgum (greater IMDs).

The smallest Q-angles were observed in 2-year-old children (node 1 in Figure 3) (*n* = 37, Q-angle mean = −3.81°, η^2^ = 0.053), while the largest Q-angle was found in 9-year-olds who were BC-LS less than always (node 15: *n* = 75, Q-angle mean = −9.49°, η^2^ = 0.018). In summary, the Q-angle was unaffected by BC practices.

### 3.3. Measurement in Transverse Plane: Tibial Torsion and Anteversion

The greatest internal TT (Figure 4) was observed in the 2-year-olds (node 1), (*n* = 37, TTA mean = 2.03°, η^2^ = 0.097). The second greatest internal TT (*n* = 35, TTA mean = 8.22°, η^2^ = 014) was observed in 6-year-old children (node 10) who were BC-LS always. The greatest external TT was observed in 8- and 9-year-old children (node 17) who were BC-LS more than very little, but less than always (*n* = 45, TTA mean = 14.69°, η^2^ = 0.011). ANOVA indicated statistical significance (F(9, 684) = 17.756, *p* < 0.001). In summary, more frequent BC contributed to greater internal TTAs.

Figure 5 shows node 13 and node 4 as the two largest AVA mean groupings. Node 13 is the highest (*n* = 20, AVA mean = 79.59°, η^2^ = 0.027), for 3-year-olds who were BC-LS for an unspecified frequency to less than average. The second-largest AVA mean (node 4; *n* = 34, AVA-mean = 78.64°, η^2^ = 0.033) in 2- to 5-year-old children who were BC-LS for specified frequencies. The 6- and 9-year-old children, BC-LS more than average to always (node 26) represent the smallest AVA (*n* = 58, AVA mean = 68.59°, η^2^ = Redundant). Node 10 is the second lowest AVA (*n* = 57, AVA mean = 69.3°, η^2^ = 0.000), in 8-year-old children who were BC-LS (more than never). Statistical significance for these terminal nodes as calculated by ANOVA (F(13, 685) = 5.075, *p* < 0.001). In summary, more frequent BC contributed to smaller AVAs.

## 4. Discussion

The findings of this study, which aimed at determining the relationships between LL angle measures and BC in Setswana-speaking children, found several significant results dependent on the age of the children. When children are BC (where always indicates the most regular BC, while the carrier is on foot, walking to town, or working) (Figure 1), the CART reveals a larger valgus TFA, which corresponds with results by Golding [4], who observed a constant valgus force produced on the LL at the knee compared to non-BC children.

In older children, neutral alignment was expected, and this was corroborated in the current study’s findings with small IMDs of the older children and larger IMDs in younger children. In younger children, BC more frequently was associated with (statistically non-significant) greater valgus distances. In older children, an opposite relationship was observed, where children who were BC less frequently had larger valgus angles, except for children always BC, the relationship was the same as in younger children. Normally, Q-angles increased with an increase in age, and this trend was also observed in BC children. The post hoc Sheffe test did not indicate any significant differences between nodes 15 and 16, nor between nodes 16 and 18 (Figure 3), thus it was assumed that Q-angles are unaffected by BC.

Secondly, the relationship between TP torsion measures (AVA and TTA), in BC and non-BC children is determined. As expected, the greatest internal TTA was observed in the youngest (2-year-olds), whilst the second largest internal TTA was observed in 6-year-old children who were always BC. This was an unexpected result, 6-year-olds normally present with neutral alignment or slight external rotation in terms of TTA. Children who were BC more frequently tended to present with larger internal TTA than those who were BC less frequently. The current findings agree with the observation of 4, where BC would contribute to valgus force at the knee and constant internal TTA force and contradicts the postulations of Knight [12] and Leveau and Bernhardt [13].

Previous studies found greater AVA in younger children that decrease with age [21,22], which was observed in the current study, where the youngest children presented with the largest AVA (Figure 5). Children BC more frequently typically present with smaller AVA than children BC less frequently, and children typically older have a lower AVA than the younger children.

The main finding of the current study is that more frequent BC practices contributed to the development of genu valgum, and not genu varum as previously postulated by Knight [12] and Leveau and Bernhardt [13]. More frequently BC children present with greater valgus TFAs, internal TTAs and decreased AVAs, while the IMD and Q-angle remain unaffected by BC practices. All the relationships mentioned between LL development and BC practices fall within the normal limits of alignment. In the event that frequent BC contributed to malalignments, such as excessive genu valgum or excessive internal tibial torsion, it may have placed higher burdens on the health system to address and treat the malalignment. If this was the case, parent education in suitable BC practices would be required.

The findings of this study should be interpreted against the limitations experienced. Firstly, the sampling was conducted in a relatively small demographic area; however, the North-West Province is widespread and further afield consists of rural tribal villages, often inaccessible. Sampling from a larger demographic area in a randomised manner would render the findings more general of developmental trends. Blank or incomplete questionnaires that had to be regarded as missing data were returned and could not be included in this analysis. The questionnaire did not collect data regarding birthing method (c-section versus vaginal birth) which could arguably influence lower limb alignments. It is suggested that future research include these aspects when investigating lower limb development of children.

## 5. Conclusions

In conclusion, results from the analysis classified relationships between BC frequency and specific dependent variables (various LL measures), depending on the correlations between variables grouped per node. BC children presented with larger genu valgum angles, smaller AVA and larger internal TTAs than non-BC counterparts. These relationships between BC children and LL development serve as an impetus for further research into the specific BC practices (duration and age until BC), to determine which, if any, of the BC practices exacerbates LL malalignments.

## Figures and Tables

**Figure 1 children-09-00263-f001:**
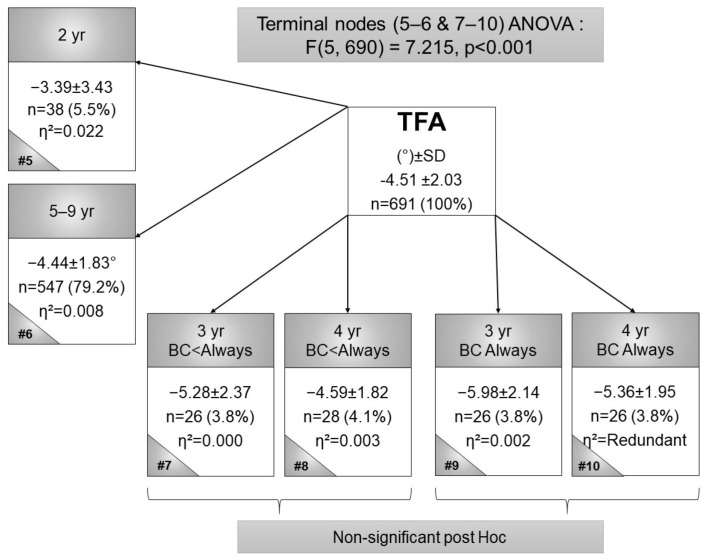
Terminal nodes CART for the tibiofemoral angle in black Setswana-speaking children back-carried with legs spread, according to age. Legend: # = node number, η^2^ = partial eta squared, BC = back-carried, TFA = tibiofemoral angle, SD = standard deviation, and yr = year.

**Figure 2 children-09-00263-f002:**
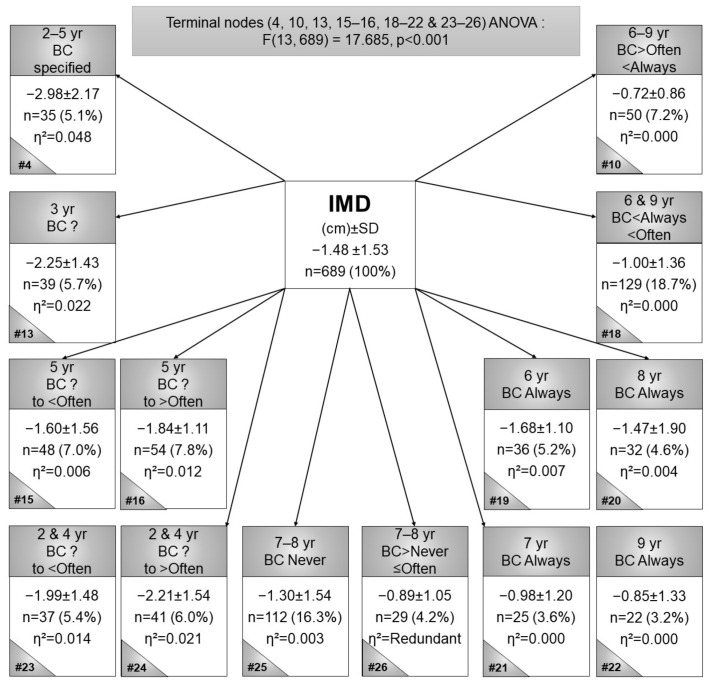
Terminal nodes CART for intermalleolar distances in black Setswana-speaking children back-carried with legs spread, according to age. Legend: # = node number, η^2^ = partial eta squared, BC = back-carried, IMD = intermalleolar distance, SD = standard deviation, and yr = year.

**Figure 3 children-09-00263-f003:**
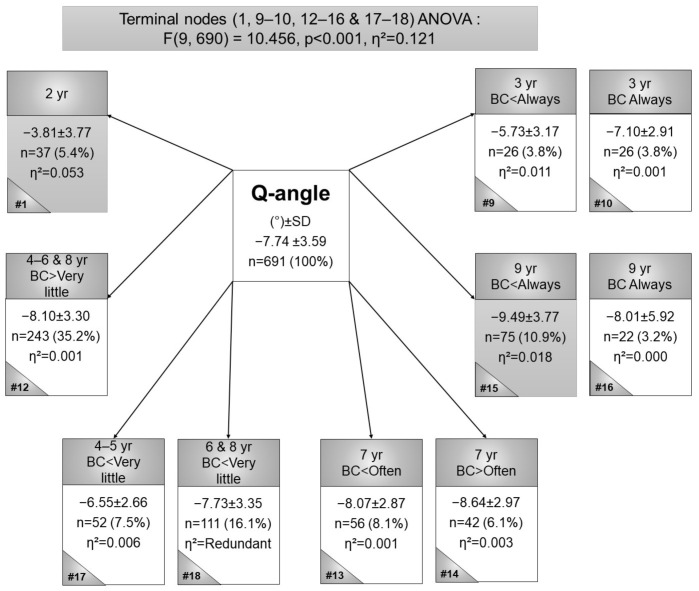
Terminal nodes CART for quadriceps angles in black Setswana-speaking children back-carried with legs spread, according to age. Legend: # = node number, η^2^ = partial eta squared, BC = back-carried, Q-angle = quadriceps angle, SD = standard deviation, and yr = year.

**Figure 4 children-09-00263-f004:**
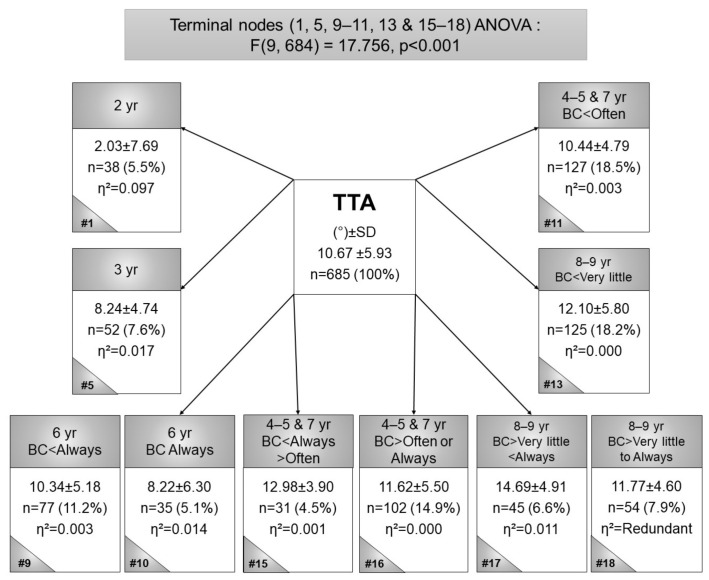
Terminal node CART for the tibial torsion angle in black Setswana-speaking children back-carried with the legs spread, according to age. Legend: # = node number, η^2^ = partial eta squared, BC = back-carried, SD = standard deviation, TTA = tibial torsion angle, yr = year.

**Figure 5 children-09-00263-f005:**
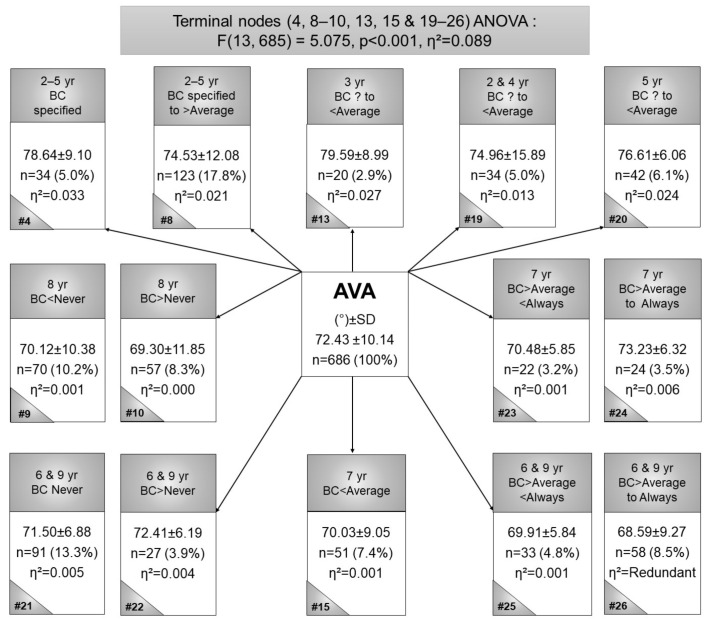
Terminal node CART for the anteversion angle in black Setswana-speaking children back-carried with the legs spread, according to age. Legend: # = node number, η^2^ = partial eta squared, AVA = anteversion angle, BC = back-carried, SD = standard deviation, yr = year.

**Table 1 children-09-00263-t001:** Descriptive statistics for frontal and transverse plane measures of the total group, per sex and age in black 2- to 9-year-olds.

	TFA			Q-Angle			IMD			AVA			TTA		

Age (years)	Total Mean (°) ± SD(*n*)	Females Mean (°) ± SD (*n*)	Males Mean (°) ± SD (*n*)	Total Mean (°) ± SD (*n*)	Females Mean (°) ± SD (*n*)	Males Mean (°) ± SD (*n*)	Total Mean (cm) ± SD (*n*)	Females Mean (cm) ± SD (*n*)	Males Mean (cm) ± SD (*n*)	Total Mean (°) ± SD (*n*)	Females Mean (°) ± SD (*n*)	Males Mean (°) ± SD (*n*)	Total Mean (°) ± SD (*n*)	Females Mean (°) ± SD (*n*)	Males Mean (°) ± SD (*n*)
2	−3.39 ± 3.43 (*n* = 38)	−4.00 ± 3.46 (*n* = 15)	−2.99 ± 3.43 (*n* = 23)	−3.81 ± 3.77 (*n* = 38)	−4.55 ± 3.32 (*n* = 15)	−3.33 ± 4.03 (*n* = 23)	−2.09 ± 1.76 (*n* = 37)	−2.94 ± 1.56 (*n* = 14)	−1.58 ± 1.70 (*n* = 23)	75.82 ± 18.87 (*n* = 37)	75.30 ± 15.23 (*n* = 14)	76.13 ± 21.10 (*n* = 23)	2.03 ± 7.69 (*n* = 37)	4.50 ± 7.09 (*n* = 14)	0.53 ± 7.79 (*n* = 23)
3	−5.63 ± 2.26 (*n* = 52)	−5.70 ± 1.83 (*n* = 24)	−4.86 ± 1.72 (*n* = 26)	−6.41 ± 3.09 (*n* = 52)	−7.08 ± 2.71 (*n* = 24)	−5.84 ± 3.31 (*n* = 28)	−2.71 ± 1.76 (*n* = 52)	−2.29 ± 1.68 (*n* = 24)	−3.08 ± 1.78 (*n* = 28)	77.61 ± 13.75 (*n* = 52)	79.53 ± 4.36 (*n* = 24)	75.96 ± 18.29 (*n* = 28)	8.24 ± 4.74 (*n* = 52)	8.38 ± 3.62 (*n* = 24)	8.13 ± 5.59 (*n* = 28)
4	−4.97 ± 1.90 (*n* = 54)	−5.01 ± 2.08 (*n* = 29)	−4.86 ± 1.72 (*n* = 26)	−7.46 ± 2.92 (*n* = 54)	−8.16 ± 3.11 (*n* = 29)	−6.65 ± 2.49 (*n* = 25)	−2.03 ± 1.28 (*n* = 53)	−2.04 ± 1.30 (*n* = 28)	−2.01 ± 1.29 (*n* = 25)	75.70 ± 7.02 (*n* = 52)	74.96 ± 3.50 (*n* = 27)	76.50 ± 9.49 (*n* = 25)	11.21 ± 3.94 (*n* = 51)	11.14 ± 3.46 (*n* = 27)	11.28 ± 4.49 (*n* = 24)
5	−4.71 ± 2.02 (*n* = 112)	−5.13 ± 2.01 (*n* = 54)	−4.36 ± 2.00 (*n* = 56)	−7.59 ± 3.40 (*n* = 112)	−8.09 ± 3.65 (*n* = 54)	−7.13 ± 3.10 (*n* = 58)	−1.84 ± 1.51 (*n* = 112)	−1.79 ± 1.34 (*n* = 54)	−1.89 ± 1.67 (*n* = 58)	75.19 ± 8.24 (*n* = 111)	76.41 ± 6.07 (*n* = 54)	74.06 ± 9.76 (*n* = 57)	11.24 ± 4.73 (*n* = 111)	11.40 ± 5.08 (*n* = 54)	11.08 ± 4.42 (*n* = 57)
6	−4.29 ± 1.61 (*n* = 113)	−4.43 ± 1.78 (*n* = 50)	−4.20 ± 1.46 (*n* = 63)	−7.96 ± 3.29 (*n* = 113)	−8.68 ± 3.56 (*n* = 50)	−7.39 ± 2.96 (*n* = 63)	−1.15 ± 1.18 (*n* = 113)	−1.22 ± 1.30 (*n* = 50)	−1.10 ± 1.09 (*n* = 63)	70.37 ± 7.96 (*n* = 113)	68.73 ± 7.09 (*n* = 50)	71.67 ± 8.42 (*n* = 63)	9.68 ± 5.62 (*n* = 112)	9.66 ± 5.83 (*n* = 50)	9.69 ± 5.48 (*n* = 62)
7	−4.40 ± 1.60 (*n* = 98)	−4.67 ± 1.56 (*n* = 51)	−4.01 ± 1.61 (*n* = 50)	−8.32 ± 2.91 (*n* = 98)	−8.44 ± 2.94 (*n* = 51)	−8.18 ± 2.91 (*n* = 47)	−1.11 ± 1.21 (*n* = 98)	−1.25 ± 1.23 (*n* = 51)	−0.95 ± 1.18 (*n* = 47)	70.92 ± 7.85 (*n* = 97)	69.91 ± 8.25 (*n* = 51)	72.05 ± 7.30 (*n* = 46)	11.17 ± 5.88 (*n* = 98)	11.11 ± 6.27 (*n* = 51)	11.24 ± 5.49 (*n* = 47)
8	−4.25 ± 1.77 (*n* = 127)	−4.53 ± 1.71 (*n* = 77)	−3.80 ± 1.73 (*n* = 47)	−7.98 ± 3.30 (*n* = 127)	−8.29 ± 3.31 (*n* = 77)	−7.49 ± 3.25 (*n* = 50)	−1.23 ± 1.63 (*n* = 127)	−1.49 ± 1.71 (*n* = 77)	−0.84 ± 1.40 (*n* = 50)	69.75 ± 11.02 (*n* = 126)	69.30 ± 9.34 (*n* = 77)	70.44 ± 13.28 (*n* = 49)	12.39 ± 5.39 (*n* = 127)	12.26 ± 5.22 (*n* = 77)	12.58 ± 5.69 (*n* = 50)
9	−4.58 ± 2.09 (*n* = 97)	−4.95 ± 1.99 (*n* = 56)	−4.14 ± 2.17 (*n* = 42)	−9.16 ± 4.36 (*n* = 97)	−9.66 ± 4.27 (*n* = 56)	−8.47 ± 4.45 (*n* = 41)	−0.95 ± 1.40 (*n* = 97)	−1.09 ± 1.44 (*n* = 56)	−0.76 ± 1.33 (*n* = 41)	70.77 ± 6.92 (*n* = 96)	71.30 ± 6.25 (*n* = 55)	70.06 ± 7.75 (*n* = 41)	12.75 ± 5.54 (*n* = 97)	12.31 ± 6.06 (*n* = 56)	13.35 ± 4.76 (*n* = 41)
Total	−4.51 ± 2.03 (*n* = 691)	−4.79 *** ± 1.95 (*n* = 356)	−4.22 *** ± 2.07 (*n* = 335)	−7.74 ± 3.59 (*n* = 691)	−8.30 *** ± 3.58 (*n* = 356)	−7.14 *** ± 3.51 (*n* = 335)	−1.48 ± 0.06 (*n* = 689)	−1.56 ± 1.51 (*n* = 354)	−1.40 ± 1.55 (*n* = 335)	72.42 ± 6.92 (*n* = 684)	72.08 ± 8.35 (*n* = 352)	72.78 ± 2.07 (*n* = 332)	10.67 ± 5.93 (*n* = 685)	10.94 ± 5.70 (*n* = 353)	10.38 ± 13.71 (*n* = 332)

Legend: AVA = anteversion angle, IMD = intermalleolar distance, Q-angle = quadriceps angle, TFA = tibiofemoral angle, TTA = tibial torsion angle, and SD = standard deviation. *** at level *p* < 0.001 after Bonferroni adjustment.

## Data Availability

The raw data are available as a Appendix A.

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
