# Peer review of "Back-Carrying in Children Is Related to Lower Limb Development†"

_children, 2022, doi:10.3390/children9020263_

Round 1

Reviewer 1 Report

Dear Authors,

This is a very interesting original article. The study is well designed according, the methods and results are well presented. The discussion is interesting and well written.

Nevertheless, I have some suggestions to improve the paper:

  1. I suggest to consider changing the title into: ,,Back-carrying in children is related to more frequent development of genu valgum’’ for example, because of the limitations of the study and that it was the main finding. The rest of the results regarding the lower limb development requires further research.
  2. I suggest adding the citation in line 95
  3. I suggest adding the initials of the researcher instead of XXX in line 134

Author Response

All comments are addressed in the attached file.

Reviewer 2 Report

In this study, the effects of carrying children on their backs (BC) were investigated. In order to determine the development of the lower limb (LL), 691 data of children aged between 2 and 9 years were analyzed. The work is interesting and I have only a few small things to comment on:

  • Table 1 presents the descriptive statistics related to the influences in the frontal as well as transversal plane - in advance I would recommend to briefly explain in 2-3 sentences something about the total cohort studied. What was the age distribution between the 2-9 years across the 691 children in detail? How many children were BC and non-BC children?
  • Note the line as well as page numbering at page 6 (Chapter 3.2) - here it starts again with line 1 and page 1, probably a formatting error due to previous landscape format
  • Chapter 3.2. under line 19 you write "...smallest valgus IMD was found in 6- to 9-year-olds (node 10), BC between often 18 and always (n=320, IMDmean=-0.72cm, η2=0.000)...."
    Comment 1: What exactly does the number of 320 refer to here? In the study, according to Table 1, a total of 435 children aged 6-9 years were analyzed with the IMD.
    Comment 2: This result refers to Figure 2, but this sentence is written directly as the first sentence after Figure 1. This was a little irritating when reading - it would be better to write a direct reference to Figure 2 and if only a transition sentence like "....The analysis of the valgus IMD is presented in Figure 2, where the smallest valgus IMD was determined at.....".
  • Line 58: Be constant - write out numbers or use numerical value: "...six- and nine year old..." or "...6- and 9 year-old..." 
